# Increased Circulating CD14+ Monocytes in Patients with Psoriatic Arthritis Presenting Impaired Apoptosis Activity

**DOI:** 10.3390/biomedicines12040775

**Published:** 2024-04-01

**Authors:** Shang-Hung Lin, Chung-Yuan Hsu, Sung-Chou Li

**Affiliations:** 1Department of Dermatology, Kaohsiung Chang Gung Memorial Hospital, Kaohsiung 833, Taiwan; hongfufu@gmail.com; 2College of Medicine, Chang Gung University, Taoyuan 333, Taiwan; chungyuango@gmail.com; 3College of Medicine, National Sun Yat-sen University, No. 70, Lianhai Road, Gushan District, Kaohsiung City 804, Taiwan; 4Division of Rheumatology, Allergy, and Immunology, Department of Internal Medicine, Kaohsiung Chang Gung Memorial Hospital, Kaohsiung 833, Taiwan; 5Department of Medical Education and Research, Kaohsiung Veterans General Hospital, No. 386, Dazhong 1st Rd, Zuoying District, Kaohsiung 813414, Taiwan; 6Department of Dental Technology, Shu-Zen Junior College of Medicine and Management, Kaohsiung 821004, Taiwan; 7Department of Nursing, Meiho University, Pingtung 912009, Taiwan

**Keywords:** CD14+ monocytes, apoptosis, cathepsin L, osteoclast, psoriatic arthritis

## Abstract

Psoriatic arthritis (PsA) is a chronic inflammatory arthritis primarily affecting peripheral and axial joints. The osteolytic effect in the damaged joint is mediated by osteoclast activation. We aimed to investigate differential gene expression in peripheral CD14+ monocytes between patients with psoriatic arthritis (*n* = 15) and healthy controls (HCs; *n* = 15). Circulating CD14+ monocytes were isolated from peripheral blood mononuclear cells using CD14+ magnetic beads. Cell apoptosis was measured via Annexin V using flow cytometry. The gene expression profiling was analyzed via microarray (available in the NCBI GEO database; accession number GSE261765), and the candidate genes were validated using PCR. The results showed a higher number of peripheral CD14+ monocytes in patients with PsA than in the HCs. By analyzing the microarray data, identifying the differentially expressed genes, and conducting pathway enrichment analysis, we found that the apoptosis signaling pathway in CD14+ cells was significantly impaired in patients with PsA compared to the HCs. Among the candidate genes in the apoptotic signaling pathway, the relative expression level of cathepsin L was confirmed to be significantly lower in the PsAs than in the HCs. We concluded that the numbers of peripheral CD14+ monocytes increased, and their apoptosis activity was impaired in patients with PsA, which could lead to enhanced macrophage maturation and osteoclast activation. The resistance of apoptotic death in peripheral CD14+ monocytes may contribute to active joint inflammation in PsA.

## 1. Introduction

Psoriatic arthritis (PsA) is a chronic inflammatory musculoskeletal disease that results in peripheral arthritis, axial spondylarthritis, enthesitis, dactylitis, and functional disabilities [1]. Approximately 30% of patients with psoriasis suffer from PsA [2]. The progression of PsA significantly decreases patients’ functional status and increases their risk of death [3].

The biomechanical strain, cutaneous inflammation of psoriasis, and metabolic and microbial factors have been reported to recruit abnormal immune cells into joint cavities in PsA. Among these risk factors, biomechanical stress on the enthesis is an important initiator of PsA [4]. The microdamage in the enthesis is capable of recruiting a large number of inflammatory cells and then activating resident immunocytes during high mechanical strain [5,6]. The pathological findings of PsA in the enthesis are composed mainly of macrophage, lymphocyte, and neutrophil infiltration. Macrophages are essential in triggering enthesitis [7]. Interleukin (IL)-23, originating from monocytes and macrophages, could induce the activation of IL-23R+Th17 lymphocytes [8,9].

The erosion of bones in PsA arises from the activity of osteoclasts [10] that are derived from precursors of the monocyte/macrophage lineage [11]. The macrophage colony-stimulating factor (M-CSF) promotes the development of osteoclast precursors (OCPs) from circulating CD14+ monocytes, and it also induces the expression of receptor activator of nuclear factor-κB (RANK) on the cell surface. The receptor activator of the NF-κB ligand (RANKL) further differentiates the OCPs into osteoclasts [12]. OCP expansion has been observed in the bone marrow of patients with PsA [3]. The presence of bone marrow edema may serve as a predictive factor for the development of erosion within a year in various inflammatory joint diseases, including PsA and reactive arthritis [13]. Magnetic resonance imaging (MRI) signals may indicate altered bone remodeling or enhancement stemming from an enriched vascular supply [14]. Our previous study showed that monocytes from patients with PsA had preferential osteoclastogenesis and osteolytic effects [15]. Elevated levels of tumor necrosis factor alpha (TNF-α), a well-known proinflammatory cytokine, have been identified in the skin and joints of individuals with psoriasis [16]. TNF-α increases the production of monocyte chemotactic protein-1 (MCP-1) from the monocyte-derived osteoclasts of patients with PsA and recruits additional OCPs into joint tissues [17].

Apoptosis, the predominant form of death in immunological cells, serves as a central regulatory feature of the immune system [18]. It is a process of programmed cell death and involves two primary pathways: the intrinsic and the extrinsic pathways [19]. Caspases are involved in the mitochondrial dysfunction of the intrinsic pathway, and TNF-α/ TNF receptor 1 (TNFR1) initiates cell shrinkage and DNA fragmentation in the extrinsic pathway [19,20]. Previous studies have shown decreased spontaneous in vitro apoptosis of peripheral monocytes in patients with rheumatoid arthritis (RA) and systemic juvenile idiopathic arthritis (JIA) [21,22].

Monocytes are the precursors of macrophages and osteoclasts, two key cells in the pathogenesis of PsA [23]. Dysregulated apoptosis in inflammatory cells has been reported in several autoimmune diseases [24,25]. To date, it remains unknown how apoptotic death in peripheral CD14+ monocytes controls inflammation in patients with PsA. We will investigate the difference in gene expression in peripheral CD14+ monocytes from patients with PsA and healthy controls (HCs).

## 2. Materials and Methods

### 2.1. Subject Enrollment

This study was approved by the Institutional Review Board of Chang Gung Memorial Hospital (IRB-201802336A3). All individuals in the PsA group met the classification for psoriatic arthritis (CASPAR) criteria for diagnosis, confirmed by both dermatologists and rheumatologists at the Department of Dermatology of Kaohsiung Chang Gung Memorial Hospital. The HCs were age- and sex-matched patients who received excisions of cutaneous benign neoplasms at our department. To ensure the absence of psoriatic lesions or inflammatory joint pain, the HCs underwent thorough examinations. Patients with active infections were excluded. All of the patients provided written informed consent.

### 2.2. Monocyte Enrichment

A standard peripheral venous blood sample was obtained from each patient, and 30 mL of whole blood was processed to eliminate the red blood cells, yielding buffy coats enriched with peripheral blood mononuclear cells (PBMCs). CD14+ monocytes were then isolated from the PBMCs using CD14+ MicroBeads (Miltenyi Biotec, Bergisch Gladbach, Germany), and the purity of the CD14+ cells post-selection was approximately 96.4%, as determined by flow cytometry analysis based on a previous study [26]. For the eight patients with severe PsA, peripheral blood was acquired at baseline and after 28 weeks of standard biological treatment (adalimumab, 40 mg by subcutaneous injection every other week, or ixekizumab, 160 mg by subcutaneous injection at week 0, followed by 80 mg every 4 weeks). Six patients with PsA received adalimumab treatment for 7 months, and the other two patients with PsA received ixekizumab treatment for 7 months. All of the other patients and the healthy controls had one sample of blood collected and processed for this study.

### 2.3. Gallium-67 Whole-Body Scans and Interpretation

Total body imaging was conducted 48 h after intravenous administration of 111 MBq (3 mCi) of gallium-67 (^67^Ga) citrate. The imaging utilized a dual-headed variable-angle gamma camera (Symbia T; Siemens Medical Solutions, Hoffman Estates, IL, USA) equipped with a medium-energy collimator. For the acquisition, the three primary energy peaks of ^67^Ga, 93 keV, 184 keV, and 300 keV were selected. The whole-body scans were performed at a rate of 8 cm/min, from head to toe.

In the visual assessment of the images, the result was deemed to be positive when the focal radiotracer accumulation in joints exhibited higher intensity than the liver uptake. Conversely, negative results were obtained when the tracer activity was confined to its normal biodistribution patterns.

All imaging studies were independently reviewed and interpreted by two experienced board-certified nuclear medicine physicians without knowledge of the patient’s identity, clinical history, or the results of other studies. In instances of discrepant or ambiguous interpretations, a third board-certified nuclear medicine specialist was consulted to facilitate consensus.

### 2.4. Microarray Assay and Analysis

CD14+ monocytes from patients with PsA and HCs were lysed in Trizol reagent (ThermoFisher Scientific, Waltham, MA, USA). Total RNA was extracted using Direct-zol™ RNA Kits (Zymo Research, Irvine, CA, USA) according to the manufacturer’s protocol. The collected RNA samples were first subjected to quality control by measuring the RNA integrity number (RIN) values with TapeStation 4200 (G2991BA, Agilent, Santa Clara, CA, USA). The RNA samples with RIN ≥ 7 were subjected to further processing using the WT PLUS reagent kit (902280, Thermo Fisher Scientific, Inc., Carlsbad, CA, USA), followed by measuring the gene expression level using a Clariom D Human microarray chip (902927, Thermo Fisher Scientific, Inc., Carlsbad, CA, USA). The generated raw CEL files were analyzed using Transcription Analysis Console 4.0 (Thermo Fisher Scientific, Inc., Carlsbad, CA, USA) to conduct ANOVA, identify the differentially expressed genes, and generate the volcano plot. In addition, we also used Partek Genomics Suite 7.0 (Partek, St. Louis, MO, USA) to conduct pathway enrichment analysis. The microarray raw data are available in the NCBI GEO database, accession number GSE261765.

### 2.5. Real-Time qPCR Assay

Quantitative real-time polymerase chain reaction (qRT-PCR) was performed on the Roche LightCycler^®^ 96 System (Roche Applied Science, Penzberg, Upper Bavaria, Germany) using the Fast SYBR Green PCR Master Mix (Applied Biosystems; Thermo Fisher Scientific, Inc., Carlsbad, CA, USA). The PCR program consisted of initial denaturation at 95 °C for 20 s, followed by 45 cycles of qRT-PCR at 95 °C for 3 s (denaturation) and 60 °C for 30 s (annealing, extension, and reading fluorescence). The primer sequences for cathepsin L (*CTSL*), inhibitor of nuclear factor kappa B kinase subunit beta (*IKBKB*), and 18S are listed in Appendix A.

### 2.6. Flow Cytometry Assay

For the evaluation of apoptotic cells, 3 × 10^5^ CD14+ monocytes/well were cultured in RPMI 1640 (Gibco, Karlsruhe, Germany) with FBS (10%, *v*/*v*; Invitrogen, Waltham, MA, USA) and 1% (*v*/*v*) penicillin/streptomycin solution (Gibco, Karlsruhe, Germany) in a 5% CO_2_ atmosphere at 37 °C. The cells were either stimulated with 100 ng/mL of TNF-α (PeproTech, Rocky Hill, NJ, USA) or 4% dimethyl sulfoxide (DMSO) (Sigma, St. Louis, MO, USA) for 24 h. The 4% DMSO induced obvious apoptosis, according to a previous study; thus, it served as the positive control group [27]. The CD14+ monocytes were washed twice with cold PBS with 1% FBS; then, we resuspended the cells in Annexin V Binding Buffer (BioLegend, San Diego, CA, USA) at a concentration of 1 × 10^6^ cells/mL. The cells were stained with Annexin V-FITC (1:50, BioLegend, San Diego, CA, USA) for 20 min at room temperature. After being washed with Annexin V Binding Buffer, the 7-AAD (Cayman, Ann Arbor, MI, USA) was also stained and incubated for 10–15 min at room temperature. The percentage of apoptotic cells was measured using flow cytometry. The apoptotic cells were defined as Annexin V-positive and 7-amino actinomycin D (AAD)-negative [28].

### 2.7. Statistical Analysis

The age, sex, number of peripheral CD14+ monocytes, percentage of apoptotic cells, and expression of *CTSL* and *IKBKB* were compared between the patients with PsA and HCs using the Mann–Whitney U test and ANOVA after establishing that they followed a normal distribution. All statistical analyses were performed with SPSS 29.0 (IBM, Armonk, NY, USA). A *p* value less than 0.05 was considered statistically significant for all tests.

## 3. Results

### 3.1. Subject Information

Fifteen patients with PsA (male/female: 8/7, average age: 47.6 years old) and fifteen HCs (male/female: 8/7, average age: 44.1 years old) were recruited (Table 1). Thirteen of the fifteen patients with PsA had severe psoriasis (average PASI of 17.6), and all fifteen had peripheral arthritis, including 46.7% with axial arthritis, 33.3% with dactylitis, and 66.7% with enthesitis. The average disease duration of PsA was 8.9 years (Table 1).

### 3.2. Higher Level of Monocytes in Patients with PsA

To determine whether there was a higher number of peripheral CD14+ monocytes in the patients with PsA (*n* = 15) than in the HCs (*n* = 15), peripheral CD14+ monocytes were obtained and sorted using CD14+ magnetic beads. The results showed a significantly higher level of CD14+ monocytes in the patients with PsA compared to the HCs (*p* < 0.001) (Figure 1A).

We then determined whether the level of CD14+ monocytes returned to the baseline level of the HCs after successful treatment. Eight of fifteen patients with PsA received biologic treatment for more than 7 months. CD14 monocytes from these eight patients achieved ACR20 after biologic treatment. The number of CD14 monocytes significantly decreased after biologic therapy (*p* = 0.01) (Figure 1B). This proved that the level of CD14+ monocytes (OCPs) from peripheral monocytes was significantly higher in patients with PsA than in HCs and that the treatment of PsA restored them to a more normal level.

### 3.3. Increased ^67^Ga Activity in the Joints of Patients with PsA

The patients with PsA could present with oligoarthritis or polyarthritis. A ^67^Ga scan was used to detect the increased white cell infiltration. To evaluate the involvement of joint inflammation, we used ^67^Ga scans to determine the inflammatory joints in the whole body. Three patients with PsA with active arthritis showed increased gallium activity in all of their tender and swollen joints. Representative images from one patient with PsA are shown in Figure 2. We could therefore evaluate the inflammatory joints in patients with PsA easily.

### 3.4. Repressed Apoptosis Pathway in the Monocytes from Patients with PsA

To investigate why the patients with PsA had a higher number of monocytes than the HCs, we conducted a microarray assay (Clariom D Human) on the monocyte samples from patients with PsA and HCs. The generated data were further analyzed using TAC 4.0 and Partek 7.0. As a result, 1621 upregulated genes and 1209 downregulated genes relative to the HCs were found in the PsA monocytes (*p* value < 0.05, Figure 3A, Appendix A). To derive their possible functions, we conducted pathway enrichment analysis on the two sets of differentially expressed genes. The analysis results are shown in Appendix A. As demonstrated in Appendix A, the apoptosis pathway was enriched in the downregulated pathway, which implied that apoptosis pathway activity was repressed in the monocytes of patients with PsA. Furthermore, 14 of the 1209 downregulated genes were involved in the apoptosis pathway (Appendix A). We selected two downregulated apoptosis-related genes, *CTSL* and *IKBKB*, for qPCR validation since they had the largest expression variations (1.54- and 1.66-fold, respectively). We measured the RNA expression level of *CTSL* and *IKBKB* in monocytes from the patients with PsA (n = 15) and HCs (n = 15). As shown in Figure 3B,C, qPCR validation confirmed that *CTSL* was downregulated in the monocytes of patients with PsA (*p* = 0.039). Although the expression of *IKBKB* was lower in monocytes from the patients with PsA than those from HCs, it did not reach statistical significance (*p* = 0.254).

### 3.5. Impaired Apoptosis Activity in PsA Monocytes with Flow Cytometry

Based on the microarray data and pathway enrichment analysis, the results showed that the apoptosis pathway was repressed in the monocyte samples from patients with PsA. We further conducted flow cytometry to validate the apoptosis activity in monocytes from patients with PsA (*n* = 5) and HCs (*n* = 5). As shown in Figure 4, the percentage of apoptotic cells was lower in the CD14+ monocytes from patients with PsA compared to the HCs (4.0 ± 0.5% versus 12.3 ± 2.9%, *p* = 0.009).

## 4. Discussion

This is the first study to investigate the role of apoptotic signaling in CD14+ monocytes in patients with PsA. Our results revealed a higher number of CD14+ monocytes in patients with PsA than in HCs. Interestingly, the number of CD14+ monocytes in the patients with PsA returned to the level of the HCs after successful biologic treatment. The ^67^Ga activity scan reflected increased leukocyte inflammation in the tender and swollen joints. The results of the microarray study showed a repressed apoptotic pathway in the CD14+ monocytes in patients with PsA. Further, we found an impaired apoptotic process with a decreased expression of *CTSL* in the CD14+ monocytes in patients with PsA.

Monocytes and macrophages play an essential role in the activation of the innate immune system. After recognition of pathogens, they release inflammatory cytokines such as TNF, IL-6, IL-1, and chemokines that activate and attract other immune cells to the sites of inflammation [29]. Moreover, a previous study has shown that OCPs derived from circulating CD14+ monocytes were markedly elevated in the peripheral blood of patients with PsA, and they were significantly reduced after treatment with anti-TNF agents [30]. In addition, granulocyte and monocyte adsorption apheresis could selectively remove monocytes and granulocytes from the blood and further provide an effective treatment choice for PsA [31]. Our results showed increased peripheral CD14+ monocytes in patients with PsA, which returned to a normal level after successful biologic treatment. The increased CD14+ monocytes in peripheral blood provided OCPs and contributed to active osteoclastogenesis and joint inflammation in PsA.

Early diagnosis is important to prevent irreversible joint damage in patients with PsA. However, there has not been a gold standard diagnostic tool. Ultrasound is a valuable tool for assessing enthesitis in PsA, but it lacks a well-reported methodology in most studies [32,33]. MRI can be used to assess inflammation and damage in joints, tendon sheaths, tendons, bone marrow edema, and entheses in patients with PsA. However, the limitations of MRI include 1. a limited number of anatomical areas per scan and 2. the exclusion of patients with claustrophobia or certain metallic implants [34]. ^67^Ga citrate and labeled leukocyte imaging are established techniques for diagnosing inflammation and infection. ^67^Ga uptake is increased within the leucocytes that accumulate in inflamed joints. A previous study indicated that ^67^Ga uptake occurs in joints affected by rheumatoid arthritis and reflects the degree of synovial inflammation [35]. Furthermore, ^67^Ga imaging can be used for the diagnosis of inflammation throughout the entire body. Our results showed an increased ^67^Ga uptake in the tender and swollen joints in patients with PsA. This reflected increased monocytes, OCPs, and macrophages in the inflammatory joints. We propose that ^67^Ga scanning can be used as a diagnostic tool for PsA.

Apoptosis is the predominant model of cell death within immunological cells, serving as a pivotal regulatory mechanism within the immune system [18]. Both the extrinsic (death receptor) pathway and the intrinsic (mitochondrial) pathway contribute to apoptosis. Beyond caspase-mediated proteolysis, additional proteases like cathepsins may participate in apoptosis regulation [36]. Annexin V, a recombinant phosphatidylserine-binding protein, serves as a robust tool for apoptosis detection due to its strong and specific interaction with phosphatidylserine residues [28]. Resistance to in vitro apoptosis has been reported to occur in human-activated monocytes [37]. Increased spontaneous IL-1β secretion and activated NF-kB signaling were reported to impair the apoptosis of monocytes in patients with RA [21]. In addition, overexpression of antiapoptotic molecules such as Fas-associated death domain-like interleukin-1β-converting enzyme-inhibitory protein (FLIP) [38] or self-sustained NF-kB activation [39] could also reinforce resistance to apoptosis in RA. These apoptotic defects further aggravate the disease of RA via the survival of proinflammatory monocytes. Our results showed a repressed apoptotic process in monocytes in patients with PsA. The percentage of apoptotic cells in monocytes in patients with PsA was lower than in HCs after TNF-α stimulus. This explains the increased number of activated monocytes in the inflammatory joints of PsA.

Cathepsins play significant roles in the physiological process of apoptosis. CTSL is one of the major lysosomal proteases responsible for lysosomal protein degradation and induction of apoptosis [40]. It has been reported to induce apoptosis via B-Myb in rotenone-treated neuron cells [41]. In addition, another study showed that high expression of CTSL induced increased spontaneous apoptosis in monocyte-derived macrophages in patients with coronary artery disease [42,43]. Our results showed lower CTSL expression in monocytes in patients with PsA than in HCs.

This study has several limitations. First, the case number was small, and the results may need to be validated by large-scale studies. Second, the recruited patients with PsA may have different co-morbidities that affect the number of monocytes (e.g., cerebrovascular disease, hypertension, etc.). Third, intrinsic limitations exist in this case–control study design, although we attempted to reduce the confounding factors. Fourth, human monocytes are divided into three major populations: classical, non-classical, and intermediate [44]. These three populations of monocytes present unique characteristics [29]. We investigated the apoptosis of all peripheral CD14+ monocytes combined in this study. The difference in the apoptotic activity in the three subpopulations of CD14+ monocytes should be studied further.

## 5. Conclusions

Our study demonstrated low expression of *CTSL* and impaired apoptosis in peripheral CD14+ monocytes in patients with PsA, which are likely to be causally associated with increased levels in these patients.

## Figures and Tables

**Figure 1 biomedicines-12-00775-f001:**
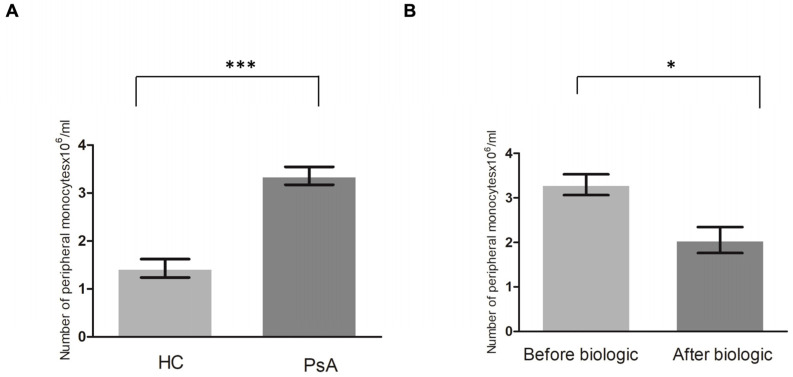
The number of peripheral CD14+ monocytes is higher in patients with PsA than in healthy controls (HCs), and it returned to the level of HCs after successful biologic treatment. (**A**) The CD14+ monocytes were analyzed from patients with PsA (*n* = 15) and HCs (*n* = 15). (**B**) To investigate the change in the number of CD14+ monocytes in patients with PsA after successful biologic treatment, the CD14+ monocytes were analyzed from eight patients with PsA before and 7 months after successful biologic treatment. * denotes *p* value < 0.05 and *** denotes *p* value < 0.001 based on Mann–Whitney U test.

**Figure 2 biomedicines-12-00775-f002:**
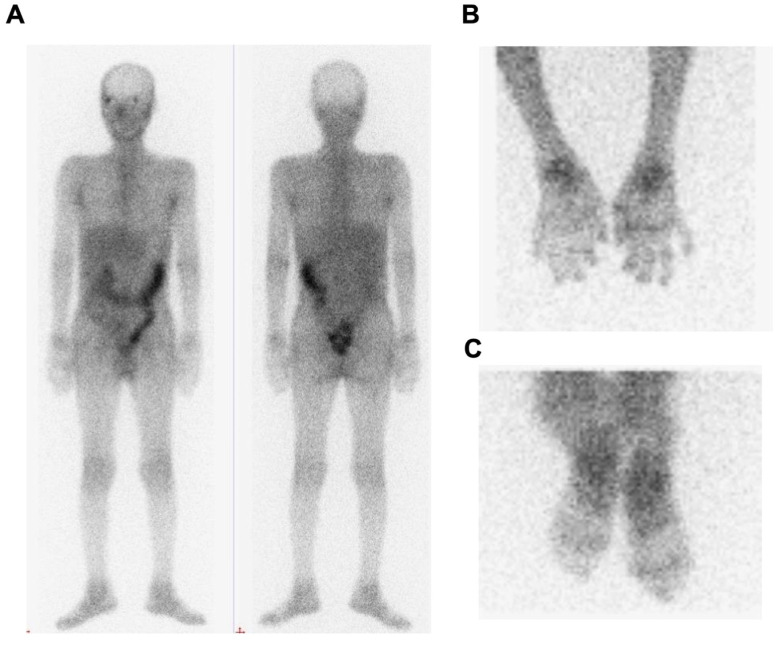
Increased activity of ^67^Ga in a patient with PsA. A 58-year-old man was diagnosed with psoriasis for 3 years. He developed psoriatic arthritis with multiple tender and swollen joints for 6 months. The ^67^Ga scan shows enhanced accumulations of the radiotracer in bilateral wrists, hands, knees, ankles, and feet (**A**). The images of bilateral hands (**B**) and bilateral feet (**C**) showed intense ^67^Ga uptake.

**Figure 3 biomedicines-12-00775-f003:**
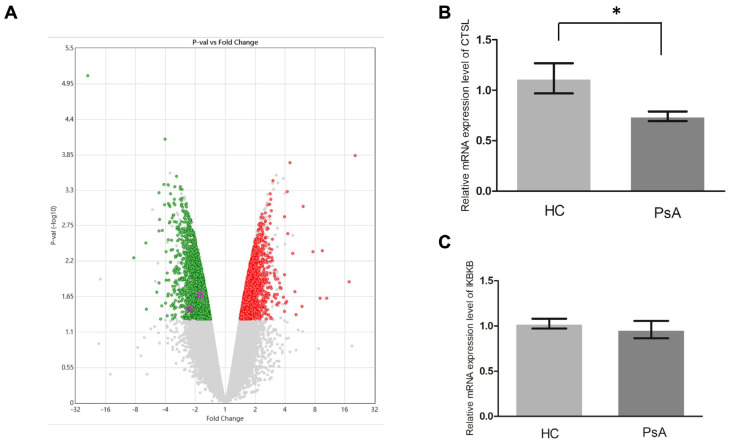
The gene expression profiles were examined using microarray and qPCR assays. (**A**) We used a volcano plot (generated using Transcription Analysis Console 4.0) to illustrate the results of the microarray assays examining the monocyte samples from two HCs and two subjects with PsA. The red and green dots denote the upregulated and downregulated genes in the PsA samples, respectively (*p* value < 0.05 based on ANOVA). (**B**,**C**) The qPCR results compared nine HC and nine PsA monocyte samples with 18S as the internal control gene. * Denotes *p* value < 0.05 based on Mann–Whitney U test.

**Figure 4 biomedicines-12-00775-f004:**
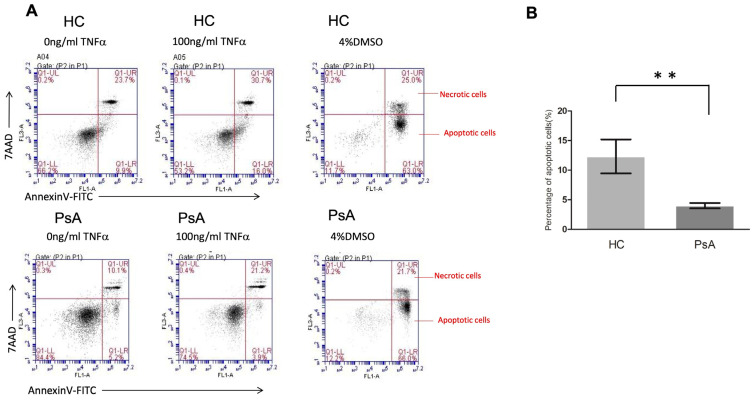
The apoptosis of CD14+ monocytes from healthy controls and patients with PsA after TNF-α treatment. CD14+ monocytes from patients with PsA and HCs were cultured with 100 ng/mL TNF-α for the evaluation of apoptosis using flow cytometry. (**A**) The apoptosis of CD14+ monocytes from patients with PsA and HCs was measured by flow cytometry. (**B**) The percentage of apoptotic cells from CD14+ monocytes of patients with PsA (*n* = 6) and HCs (*n* = 6) was measured. ** Denotes *p* value < 0.01 based on Mann–Whitney U test.

**Table 1 biomedicines-12-00775-t001:** Demographic and clinical characteristics of patients with psoriatic arthritis (PsA) and healthy controls.

	PsA (*n*= 15)	Healthy Control (*n*= 15)
Age (years)	43.7 ± 14.5	44.9 ± 12.6
Female sex no. (%)	7 (46.7)	7 (46.7)
Psoriasis (years)	17.5 ± 9.7	
Psoriatic arthritis (years)	8.9 ± 6.9	
PASI	17.6 ± 9.3	
Peripheral arthritis	15 (100)	
Peripheral and axil arthritis	7 (46.7)	
Dactylitis	5 (33.3)	
Enthesitis	10 (66.7)	
Tender joint count (of 68 joints)	16.9 ± 11.9	
Swollen joint count (of 66 joints)	9.7 ± 6.9	

PsA: psoriatic arthritis; PASI: Psoriasis Area and Severity Index.

## Data Availability

Please contact Sung-Chou Li (raymond.pinus@gmail.com) for data requests. The microarray raw data are available in the NCBI GEO database, accession number GSE261765.

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
