# Peer review of "Increased Circulating CD14+ Monocytes in Patients with Psoriatic Arthritis Presenting Impaired Apoptosis Activity"

_biomedicines, 2024, doi:10.3390/biomedicines12040775_

Round 1

Reviewer 1 Report

Comments and Suggestions for Authors

The work of the authors is interesting but I have some points to raise:

- you should expand the part of the discussion regarding the limitations of the study, acknowledging the intrinsic limitations of a case-control study design

- you could expand more in the discussion the implications of the results from a clinical perspective, thus answering the question from a clinician reader: how should or will these results impact my practice?

- in the methods, did you use ANOVA and t-test directly or did you check first for normality? You should add this in the methods, because if the data are not normally distributed the used tests are not the right ones to apply

- in the methods please report also the software and the packages you used to make the analysis and create the graphics

Author Response

Thanks for the efforts and comments from the reviewers. We have revised the manuscript accordingly. Please see our point to point responses in the attached file.  

Reviewer 2 Report

Comments and Suggestions for Authors

The authors reported about increased numbers of circulating monocytes and impaired apoptosis in patients with AS.

Comments

1.      Lines 17-18: The affected joints should be described.

2.      Line 19, 21-22, 47-49, 76-77, 102-104; 148-149, 153-154, 186, : This sentences are not clear. They should be clarified.

3.      Line 23: The language of the paper should be improved.

4.      Lines 26 and 28, 92-97 and 108-112,219-220 and 234-235 : All the repeated sentences should be corrected.

5.      References should be arranged in accordance with the Journal requirements.

6.      Introduction should be rewritten due to lack of logic (For example, lines 58-62).

7.      Line 91-92: Please, add CD14+ markers.

8.      Line 124: The role of TNF should be specified.

9.      Line 127’ 137: All abbreviations should be disclosed on first use.

10.  Table 1 should be placed in the Results section.

11.  Line 154: It is not clear what are these patients come from? This should be clarified.

12.  Fig 1: The treatment and follow-up details should be described in methods section. It is also not clear ml of what body liquid was used for reference.

13.  Section 3.3: The gallium-67 scan was not described in Methods section. This should be corrected.

14.  Section 3.4: The authors should preset more evidence for enrichment of apoptosis related gene expression.

Comments on the Quality of English Language

1.      Line 19, 21-22, 47-49, 76-77, 102-104; 148-149, 153-154, 186, : This sentences are not clear. They should be clarified.

2.      Line 23: The language of the paper should be improved.

Overall: The results are not convincing.

Author Response

(The authors gave the same response as above.)

Reviewer 3 Report

Comments and Suggestions for Authors

The manuscript regarding the role of CD14 monocytes in psoriasis is clinically relevant; however, several issues require attention in the manuscript. The first issue refers to an increased number of CD14 cells but does not specify if the increase would be of CD14CD16 positive cells as expected. The subpopulation of CD14CD16-positive monocyte is crucial see 10.3390/ijms20020291

Why were the patients were submitted to Gallium 67. What number is the ethical approval?

Figure 3 is not informative; what genes are up and which are downregulated? Part B and C are useless unless there is a description of the main genes involved.In Figure 4, the number of dead cells is very high in control at basal levels. In addition in DMSO treatment, cell death occurs through a different mechanism so the clear control should be TNF addition to the cells in culture.

The discussion and conclusions should be modified since there is no clear mechanism proposed for the increased amount of monocyte cell death in the manuscript.

Comments on the Quality of English Language

Minor grammatical mistakes.

Author Response

(The authors gave the same response as above.)

Round 2

Reviewer 2 Report

Comments and Suggestions for Authors

The authors carelessly responded the previous Comments.

Previous Comments ## 3, 10, 11, and 12 were not addressed.

Additional Comments:

1.     The language of the WHOLE paper should be significantly improved. The authors should use help of the professional proofreading agency as the text of the manuscript is not clear. This is specifically related to the Abstract, line 59-62, 119,143-145, 164-169, 177,182-186.

2.     Line 72-77: Patient treatments, dosage duration etc should be described in detail.

3.     Line 97: RNA of good quality is considered to have RIN>8. This should be corrected.

4.     Lines 120-122: the detailed protocol for treatment of cells with TNFa including concentrations should be presented.

5.     Line 131-134: Software versions and origin should be indicated.

6.     Line 138 the exact number of patients should be indicated.

7.     Lines 161-162 should be moved to Methods section.

8.     Lines213-220 are related to the Abstract. They should be removed from Discussion.

9.     Lines 234-247 are related to Introduction and should be moved there.

Comments on the Quality of English Language

1.     The language of the WHOLE paper should be significantly improved. The authors should use help of the professional proofreading agency as the text of the manuscript is not clear. This is specifically related to the Abstract, line 59-62, 119,143-145, 164-169, 177,182-186.

Author Response

Thanks for your efforts and comments. We have revised the manuscript accordingly. 

Reviewer 3 Report

Comments and Suggestions for Authors

The authors responded to most of the queries. However, I would like to point out that the limitations of the study should be under a different remark. The data of supplemental tables 1 and 2 are crude data without bioinformatic analysis and it should be added in the text.

Comments on the Quality of English Language

MInor grammatical mistakes were encountered

Author Response

(The authors gave the same response as above.)

Round 3

Reviewer 2 Report

Comments and Suggestions for Authors

The manuscript was significantly improved by the authors, however, some corrections are still required.

Comments

1.      Lines 20-21; 32-35; 241-242: These sentences are not clear. They should be rephrased.

2.       Lines 72-79: The authors should indicate here the number of subjects entrolled in each group

3.      Line 111: The protocol for RNA isolation should be presented here.

4.      Lines 11-120: Every piece of equipment should contain the information on the name of Company, City and State of purchase.

5.      Line 112: G2991BA is correct. Please, correct.

6.      Section 2.6: As the number of patients were less than 30 the authors should MannWhitney U test for statistical analyses. Therefore, the authors should recalculate their results (Lines 182, 219 233).

7.      Fig 2: A, B, and C designations should be described in the Figure caption.

8.      Line 196: The authors should indicate here what kind of microarray assay they used.

9.      References should be arranged according to the Biomedicines requirements.

Comments on the Quality of English Language

Lines 20-21; 32-35; 241-242: These sentences are not clear. They should be rephrased.

Author Response

Thanks for your efforts and comments. We have revised the manuscript accordingly. Please see our point to point responses below.  

Reviewer 3 Report

Comments and Suggestions for Authors

The manuscript is suitable for publication

Author Response

Thanks for your efforts and comments. 

Round 4

Reviewer 2 Report

Comments and Suggestions for Authors

I have no more comments. Accept as is.